# Contrasting Phenomenology of NMR Shifts in Cuprate Superconductors

**Jürgen Haase \*, Michael Jurkutat and Jonas Kohlrautz**

Faculty of Physics and Earth Sciences, University of Leipzig, Linnéstr. 5, 04103 Leipzig, Germany; jurkutat@physik.uni-leipzig.de (M.J.); kohlrautz@physik.uni-leipzig.de (J.K.)
\* Correspondence: j.haase@physik.uni-leipzig.de; Tel.: +49-341-97-32600

**Abstract:** Nuclear magnetic resonance (NMR) shifts, if stripped of their uncertainties, must hold key information about the electronic fluid in the cuprates. The early shift interpretation that favored a single-fluid scenario will be reviewed, as well as recent experiments that reported its failure. Thereafter, based on literature shift data for planar Cu, a contrasting shift phenomenology for cuprate superconductors is developed, which is very different from the early view while being in agreement with all published data. For example, it will be shown that the hyperfine scenario used up to now is inadequate as a large isotropic shift component is discovered. Furthermore, the changes of the temperature dependences of the shifts above and below the superconducting transitions temperature proceed according to a few rules that were not discussed before. It appears that there can be substantial spin shift at the lowest temperature if the magnetic field is perpendicular to the $CuO_2$ plane, which points to a localization of spin in the $3d(x^2 - y^2)$ orbital. A simple model is presented based on the most fundamental findings. The analysis must have new consequences for theory of the cuprates.

**Keywords:** superconductivity; cuprates; NMR; shift

## 1. Introduction

Soon after the discovery of cuprate high-temperature superconductors [1], the search for the understanding of their chemical and electronic properties with nuclear magnetic resonance (NMR) began, see e.g., [2–6]. NMR is a powerful local probe that measures the electron–nucleus interaction at particular nuclear sites through its effect on the nuclear levels. Thus, one has access to high and low energy properties at the atomic level of resolution as a function of temperature, pressure, and magnetic field [7]. In particular, NMR gives access to the electronic susceptibility also below the critical temperature of superconductivity ($T_c$) through shift and relaxation measurements [8].

For ordinary metals, it was known long ago, even well before the advent of NMR, that the high electronic density of states at the Fermi surface will cause special nuclear relaxation [9] in metals, which is closely related to the later observed spin shift (Knight shift [10]) in metals [11]. Indeed, for Fermi liquids that become classical superconductors with spin-singlet pairing, these spin moments vanish and lead to the disappearance of nuclear shift (Yosida function) [12] (cf. Figure 1) and relaxation (the Hebel–Slichter peak constituted the first proof of the Bardeen–Cooper–Schrieffer (BCS) theory that predicted coherence peaks just below $T_c$) [13].

The superconducting cuprates descend from antiferromagnets by doping, and this raised the question of how the Cu electronic spin will engage in conductivity and superconductivity, and how this can be monitored with NMR, as a function of doping and temperature.

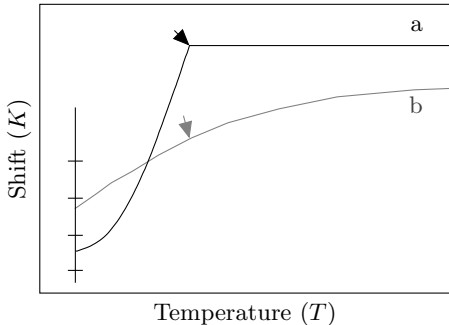

**Figure 1.** Schematic temperature (*T*) dependences of cuprate NMR shifts (*K*). a: Fermi liquid-like behavior: a temperature independent shift above the critical temperature of superconductivity ($T_c$, marked by arrow) vanishes rapidly below $T_c$ for spin-singlet pairing [12]; b: typical shift in the pseudogap range of cuprates: the shift falls with decreasing *T* already above $T_c$ and continues to drop below it. The $T \to 0$ limit is often difficult to determine experimentally.

It was soon established that the highly doped materials show nearly Fermi liquid-like behavior and that the pairing appears to favor spin singlet formation. However, at lower doping, but still above $T_c$, shifts and relaxation showed pronounced differences to a Fermi liquid. This behavior was believed to be due to the opening of a spingap, as the pseudogap was called when it was discovered with NMR [14]. The NMR hallmark of the pseudogap is the drop in Knight shift with decreasing temperature above $T_c$, cf. Figure 1.

It has to be pointed out that NMR of the cuprates is quite complex for a number of reasons.

First, the presence of crystallographically inequivalent sites in the large unit cells can cause different, overlapping resonance lines even for a single isotope.

Second, the presence of symmetry disturbing dopants leads to structural inhomogeneities that can cause large NMR linewidths. This is true in particular for nuclei with spin greater than 1/2 ($I > 1/2$) through electric quadrupole interaction, thus, also for the $^{63,65}$Cu ($I = 3/2$) and $^{17}$O ($I = 5/2$) nuclei in the $CuO_2$ plane.

Third, and difficult to distinguish from chemical inhomogeneity, there is electronic inhomogeneity that was predicted and detected early on, e.g., [15–18]. While present in all cuprates [19–23], it is not well understood so that its consequences for the various experimental probes are not clear. In NMR, one finds various degrees of line broadening in the various cuprates (see e.g., [24]) typically caused by concomitant charge and spin density variations [25]. The results cannot be dismissed as being due to sample quality, since even chemically highly ordered systems can show tremendous electronic inhomogeneity [26–28]. More recently, as will be addressed below, even the magnetic field was shown to influence the NMR splittings [29] and broadening [30]. Nevertheless, the average shift appears to be an important and reliable parameter.

Fourth, a limited penetration depth in the normal and superconducting state leads to uncertainties and signal-to-noise problems, and, finally, a partial diamagnetism in the mixed state below $T_c$ complicates NMR shift referencing. These and other complications are reasons for slow experimental progress, and they left uncertainties in the interpretation of the data.

Here, we will be concerned with shift analysis only since understanding relaxation involves a spectral range of fluctuations that may include not only the magnetic field, but also the electric field gradient. The simple uniform magnetic response, on the other hand, is responsible for the shifts for which quadrupolar effects can be eliminated much more easily. In addition, since Cu NMR does not require isotope exchange, which, for $^{17}$O, always has the risk of changing the doping level. We will focus on the most abundantly reported Cu shifts here and only discuss other shifts occasionally.

We will begin with remarks of concern for shift measurements, which help clarify shift referencing and other uncertainties. Then, we will repeat the early shift phenomenology before summarizing recent findings regarding the failure of the single-component description, which leads us to introducing

a new shift phenomenology based on the available literature data. In this phenomenology, we will highlight special observations in the shifts that question a number of earlier conclusions and hopefully help theories to advance the understanding of the electronic properties of the cuprates.

## 2. Frequencies, Shifts, and Splittings

In the most simple NMR experiment, the center of gravity of the Fourier transform of a nuclear free induction decay after a $\pi/2$ pulse determines the resonance frequency ($\nu$). Often, $\nu$ coincides with the peak of the resonance line, which serves as measure of the shift. For this to be true, the radio frequency (RF) pulse excitation should be large compared to the width of a resonance line, but this is often not the case in the cuprates, not even for a single magnetic transition. Fortunately, the broadening is mostly inhomogeneous from short and long-range shift variations [25] and one employs selective techniques (such as frequency stepping) to map out the spectral distribution of resonances approximately in lengthy experiments. Large and often temperature dependent linewidths hamper the exact determination of shifts, especially for the non-stoichiometric cuprates, and decrease NMR sensitivity. These are reasons why the lineshapes for the determination of the shifts are not always reported. Nevertheless, it appears that the Cu frequencies are quite reliable.

Loss of the NMR signal in the cuprates is of serious concern. In the superconducting state, RF penetration becomes exceedingly small, while above $T_c$, it is only limited by the anisotropic normal state conductivity. This favors the use of (oriented) powders that, on the other hand, are less favorable for orientation dependent studies. In addition, some underdoped cuprates show a sudden signal loss at some critical temperature similar to what was observed for spin glasses [31]. Thus, one has to be aware that NMR could miss important signals. Signal intensity issues have not been properly addressed with most experiments adding to uncertainties. Perhaps the strongest evidence, however, that the observed NMR signals do represent the bulk properties is the fact that the quadrupole splittings measured with NMR account for the average chemical doping [32–34].

For reasons of symmetry, the crystal *c*-axis coincides with that of the local magnetic field at the nuclei if the external field ($B_0$) is parallel to $c$ ($c \parallel B_0$). This must be correct for orbital as well as spin shift tensors ($K_{cc}$). Likewise, the largest principle component of the electric field gradient at the Cu nucleus must point in the same *c*-direction ($V_{cc}$). Given a rather small or vanishing orthorhombic distortion, the direction of the other two principle axes of all of the tensors are not known with certainty, but the asymmetry of all tensors is expected to be rather small. For planar oxygen nuclei, the situation is very different, as the direction of the $\sigma$-bond in the plane dominates local symmetry and the asymmetry perpendicular to the $\sigma$-bond is expected to be much larger (in and out of the plane).

For $^{63,65}$Cu with $I = 3/2$, we expect three resonance lines due to the quadrupole interaction. With the field along the high symmetry axis ($c \parallel B_0$), one has in the leading order for a dominating Zeeman term,

$$\nu_{\parallel,n} = \nu_0 + n\nu_Q, \tag{1}$$

where $n = 0, \pm 1$ denotes the $2I = 3$ transitions of the $^{63,65}$Cu nucleus and $\nu_Q$ is the quadrupole splitting, cf. Figure 2. Any magnetic frequency shift affects all transitions the same way so that we can discuss all shifts just for the central ($n = 0$, $\nu_0$) transition, cf. Figure 2.

If the magnetic field is perpendicular to the crystal *c*-axis ($c \perp B_0$), the central transition ($\nu_{\perp,0}$) is affected by quadrupole related shifts in higher order. However, if $\nu_Q$ has been determined from measurements for $c \parallel B_0$, the magnetic shift can be calculated from $\nu_{\perp,0}$ (for given tensor orientation).

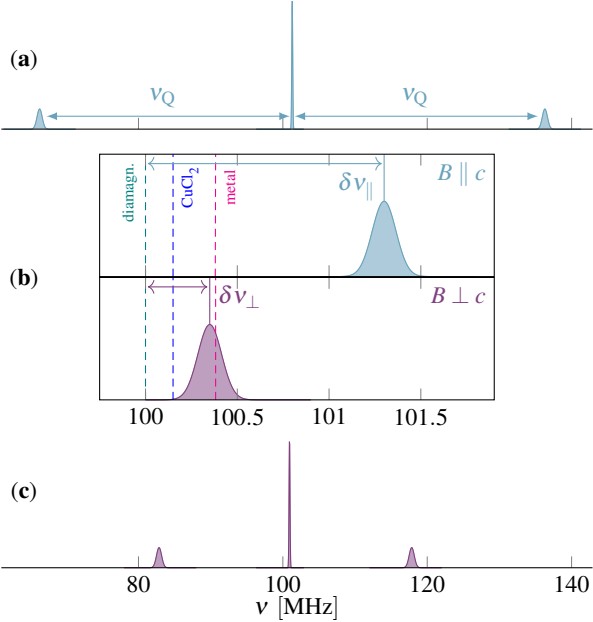

**Figure 2.** Illustration of planar Cu ($I = 3/2$) spectra in the cuprates. (**a**) For $c \parallel B_0$, the splitting between the central and the two satellite lines defines the quadrupole frequency $\nu_Q$ (a slight distribution in $\nu_Q$ affects the widths of the satellites, but not the central transition), cf. (1). (**c**) Splitting for $c \perp B_0$ is nearly half that in $c \parallel B_0$ for a vanishing asymmetry of the field gradient tensor, but the central transition is shifted by about 600 kHz due to higher order quadrupole effects. (**b**) Blowup of the central region shows the central transitions for both directions of the field after the higher order effects have been subtracted for $c \perp B_0$; the resonance frequencies of different reference compounds are indicated, cf. (2).

For the discussion of the Cu shift, we can focus on $\nu_0$ in (1) that is measured with respect to a reference material resonating under identical conditions in the same field ($B_0$) at $\nu_{ref}$, so that the NMR shift in frequency units is given by,

$$\delta\nu = \nu_0 - \nu_{ref}. \tag{2}$$

This NMR frequency shift is influenced by the diamagnetic shielding contribution from the core electrons (which is similar for all materials) and a paramagnetic Van Vleck term due to mixing with excited states depending on the chemical structure. The latter is the determining factor for the importance of the chemical shift. Electronic spin effects are expected to alter the NMR resonance frequency [7,35], in particular since the hyperfine coupling amplifies spin effects over those from orbital motion of free carriers (in the absence of strong spin-orbit coupling). Obviously, one would like to use as reference a material that has a shift dominated by the core electrons only. Then, the (isotropic) shifts $\delta\nu$ relate to interesting effects (Van Vleck and spin effects). Unfortunately, shift referencing in the cuprates varies, i.e., most of the early published NMR work relied on CuCl as a reference, a material that was later shown to have a significant Van Vleck contribution ($\delta\nu/\nu_{ref} = -0.15\%$) [36]. In order to use the magnitude of the NMR shift to relate to other probes or theory, one has to make sure that the shifts are properly referenced. All shifts shown here have been corrected where possible, cf. also Appendix A.

Since one expects the mentioned frequency shifts to be proportional to the magnetic field [7], one typically uses relative shift values that are proportional to the uniform susceptibilities (spin and orbital),

$$K = \delta\nu/\nu_{ref} \equiv K_L + K_s, \tag{3}$$

where we defined orbital ($K_L$) and spin ($K_s$) shift contributions.

Currently, it is not fully known whether the shifts in the cuprates are indeed proportional to the field. There have been reports of a field dependence of $K$ [37] in high fields, and high fields were shown

to induce charge density variations [38] as well. Even moderate fields were shown to be able to induce distributions of quadrupolar splittings [30]. However, some field dependent measurements have been performed and nuclear quadrupole resonance (NQR) experiments ($B_0 = 0$) show the absence of significant static magnetism. The study of the field dependence of the shifts is problematic since, at low magnetic fields, the rather large quadrupole interaction for Cu makes precise magnetic shift measurements difficult, apart from lowering the NMR intensity that is nearly quadratic in the field. We stipulate that the field dependence of $K$ at lower fields ($\approx$10 T) can be neglected.

If one expects macroscopic shielding effects, an internal shift reference is to be used. This can be rather difficult, but comparison with the NMR shift of other nuclei within the same sample can be helpful. This has been employed rarely [27,39–41], rather, the diamagnetism in the mixed state was estimated based on phenomenological theory. This adds some uncertainty to the NMR shifts at low temperatures in the typical fields used (the distance between fluxoids is small compared to the penetration depth, which makes the overall field variation small in terms of shift and linewidth effects).

Lastly, one needs to mention the availability of single crystals. Since large enough crystals were rarely available, stacks of small crystals or *c*-axis aligned powders were used for shift measurements. This makes measurements in the *a*-*b*-plane difficult, e.g., if there was a sizable asymmetry in the plane. In addition, if one measures the shifts with the magnetic field in the CuO$_2$ plane, higher order quadrupole effects are important and both tensors' principal axes do not necessarily share the same directions in the plane. Unfortunately, shift differences upon rotating the magnetic field in the CuO$_2$ plane are sparse.

Considering all the effects mentioned above, $^{63,65}$Cu NMR is quite reliable if the linewidth is not exceedingly large. The shifts vary over a large range as a function of temperature and doping so that partial diamagnetism is less important. Nevertheless, one has to be aware of discrepancies.

## 3. Early Shift Phenomenology

In early shift experiments, the decrease of shift for $c \perp B_0$ at low temperatures was interpreted as evidence for spin singlet pairing, and the decrease of shift above $T_c$ marked the discovery of the pseudogap. A more detailed picture was discussed in particular based on data from YBa$_2$Cu$_3$O$_{6+y}$ aligned powders and single crystals [5,6,35,42,43].

For this important family, the underdoped materials, cf. Figure 3 (left panel), show pseudogap behavior for $K_\perp$, but $K_\parallel$ is rather temperature independent; the stoichiometric, slightly overdoped material shows a temperature independent $K_\perp$ above about $T_c$, cf. Figure 3 (second from left), below $T_c$, the shift decreases; for $c \parallel B_0$, after the diamagnetic correction, the shift appeared almost unchanged. Figure 3 also shows quite different temperature dependent shifts observed in other cuprates. La$_{2-x}$Sr$_x$CuO$_4$ exhibits Fermi liquid-like behavior for $K_\perp$ and no temperature dependence for $K_\parallel$. The situation is again different for HgBa$_2$CuO$_{4+\delta}$, where there is a sizable shift also present for $c \parallel B_0$.

The shifts remaining at the lowest temperatures in both directions of the magnetic field (with regard to the crystal *c*-axis) were believed to be orbital shifts. The then determined orbital shift anisotropy was close to what one expected for an isolated Cu$^{2+}$ ion in the square planar arrangement of the CuO$_2$ plane. Similarly, the quadrupole splitting was in support of such a Cu$^{2+}$ state [35]. These findings entail the presence of a nearly half filled $3d(x^2 - y^2)$ orbital. As a consequence, one expected a negative spin shift from core polarization and dipolar effects for $c \parallel B_0$ ($A_\parallel$), but a much smaller and positive shift for $c \perp B_0$ ($A_\perp$). This can be reliably estimated but is also known from experiments on other such Cu$^{2+}$ ions [35,44]. However, this was not observed, cf. Figure 3, and it was concluded that a more complicated hyperfine scenario must be correct, one that involves also an isotropic term $B$,

$$K_\alpha = A_\alpha \chi_{Cu} + 4B\chi_O, \tag{4}$$

where the two susceptibilities are from Cu and O spins, respectively. $A_\alpha$ denotes the anisotropic onsite hyperfine constant and $B$ the so-called transferred hyperfine term.

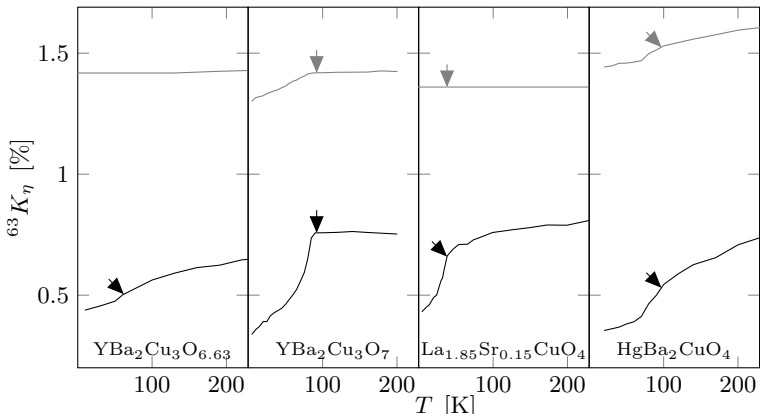

**Figure 3.** $^{63}$Cu shift for four different materials with the magnetic field ($B_0$) parallel ($K_\parallel$, upper) and perpendicular ($K_\perp$, lower) to the crystal *c*-axis vs. temperature. Arrows denote the reported $T_c$ (references are given in the Appendix A), and, for further discussion, see the text. Doping levels are near optimal doping, except for YBa$_2$Cu$_3$O$_{6.63}$.

It was noted early on that the Cu$^{2+}$ spin may exchange through planar O and thus involve the Cu 4*s* state, which should lead to an isotropic shift term [45] that could explain the data in terms of a single fluid, rather than in terms of Cu and O spins [46]. This can be tested with NMR since a single electronic fluid has a single susceptibility. If one defines the change of the spin susceptibility $\chi_s$ between any two temperatures $T_k$ and $T_l$ by $\Delta\chi_s \equiv \chi_s(T_k) - \chi_s(T_l)$, proportional shift changes $\Delta K_s \equiv K_s(T_k) - K_s(T_l)$ at all nuclei should be found,

$$\Delta K_{j,s} = h_j \Delta\chi_s, \tag{5}$$

if $j$ denotes a particular isotope or lattice site and $h_j$ the effective hyperfine constant. This is different from what one would expect if (4) were applicable, with different temperature dependences for $\chi_{Cu}$ and $\chi_O$ (and $A_\alpha \neq 4B$).

With experiments on YBa$_2$Cu$_3$O$_{6.63}$, it was concluded from shifts for Cu and O in the plane that a single susceptibility is at work [43], cf. Figure 4a, in strong support of the single fluid scenario. In a second set of experiments on YBa$_2$Cu$_4$O$_8$ [47], cf. Figure 4b, the same conclusion was reached, although a slight deviation could be seen that was still, however, within the error bars. Note that there is no temperature dependent shift for Cu and $c \parallel B_0$. These findings influenced theory considerably, although doubts of a single-band scenario persisted [48–51].

The adopted single fluid picture had to explain the absence of the spin shift for $c \parallel B_0$ through an accidental cancellation of the hyperfine terms, i.e.,

$$K_\alpha = [A_\alpha + 4B]\chi_s, \tag{6}$$

with $A_\parallel + 4B = 0$. While questioned, this coincidence was widely accepted, e.g., [52], even for other families, such as La$_{2-x}$Sr$_x$CuO$_4$, cf. Figure 3. Other materials showed temperature dependent shifts also in $c \parallel B_0$, which pointed to a violation of the accidental cancellation in these systems. However, for HgBa$_2$CuO$_{4+\delta}$ and near room temperature, cf. Figure 3, we can have $K_\parallel/K_\perp \approx 0.5$, which requires a rather large change in the hyperfine scenario.

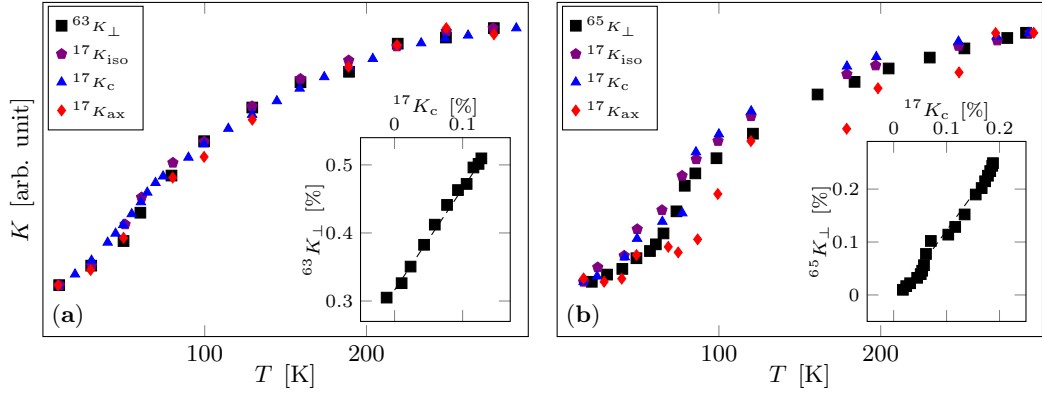

**Figure 4.** Proof of single fluid model for $YBa_2Cu_3O_{6.63}$ (**a**) (data from [43]). The temperature (*T*) dependences of the shifts for $^{63}Cu$ and $^{17}O$ are very similar. Proportional changes for $^{63}K_\perp$ and $^{17}K_c$ lead to a straight line in the inset, cf. (5). Proof of single fluid model for $YBa_2Cu_4O_8$ (**b**) (data from [47]). The temperature dependences of the shifts for $^{65}Cu$ and $^{17}O$ are similar. The proportionality in the changes of $^{65}K_\perp$ and $^{17}K_c$, cf. (5), is not perfect.

## 4. Failure of the Single Fluid Model

$La_{2-x}Sr_xCuO_4$ was assumed to be governed by a single fluid, but its planar Cu and O shifts showed apparently different temperature dependences (cf. Figure 5). NMR shifts, if stripped of uncertainties such as the the diamagnetic response below $T_c$, must hold definite answers. With a series of shift measurements on $La_{1.85}Sr_{0.15}CuO_4$ by using the apical oxygen NMR signal as an internal shift reference, it was shown beyond a doubt that this material violates the single fluid scenario and that two susceptibilities with different temperature dependences are required for the understanding of the data [40] (cf. Figure 5). Then, similar to (4), one has to describe the shifts by

$$K_\alpha = p_\alpha \chi_1 + q_\alpha \chi_2,$$
$$K_\alpha = p_\alpha \{\chi_{11} + \chi_{12}\} + q_\alpha \{\chi_{22} + \chi_{12}\},$$

(7)

where $p_\alpha, q_\alpha$ are effective hyperfine constants for the coupling to the two electronic spin components. Note that a third susceptibility must be introduced ($\chi_{12} = \chi_{21}$) since a coupling between any two spin components with susceptibilities $\chi_{11}$ and $\chi_{22}$ will lead to a coupling susceptibility ($\chi_{12}$) where the field at component 2 induces a polarization of component 1. It was found that the two susceptibilities discernible with NMR experiments ($\chi_1$ and $\chi_2$) showed rather different temperature dependences. One susceptibility (that dominates the Cu shifts) shows a Fermi liquid-like behavior in that it is temperature independent above about $T_c$, but falls off rapidly as the temperature is lowered, while the other susceptibility (that dominates the planar O shift) is temperature dependent already above $T_c$ and must be related to the pseudogap feature (the hyperfine coefficients $p_\alpha, q_\alpha$ are given in [40]).

While these experiments showed that the single fluid picture is not universal, it did not explain why other systems could be understood with one susceptibility, in particular the underdoped stoichiometric compound $YBa_2Cu_4O_8$ that is known to have very narrow NMR lines, which makes it a benchmark system. Within the error bars of early NMR, cf. Figure 4b, a sudden change near $T_c$ is indicated, which raises the question how one could further investigate this compound. Obviously, the application of pressure, strong enough to change the electronic properties slightly, but weak enough to leave the chemical structure unscathed, is a desirable tool. Such experiments, based on a new high-pressure NMR cell design [53], could be performed recently. Since Cu has strong orbital shifts, $^{17}O$ NMR was employed as the oxygen orbital shift is expected to be rather weak on general grounds. As a function of pressure (that is known to raise $T_c$), it was found that a two component scenario was necessary to understand the change in the shift [54]. With increasing pressure, a Fermi

liquid-like component appears in the data (amplified by a factor of about 9 at about 6 GPa compared to the ambient pressure data). Thus, this benchmark system weakly breaks the single-fluid picture at ambient pressure, but fails to behave as such at higher pressure. These experiments showed the failure of a single-fluid description in another family of materials, while agreeing with previously published accounts [47]. Thus, it appeared that the single-fluid picture emerging through NMR was rather accidental.

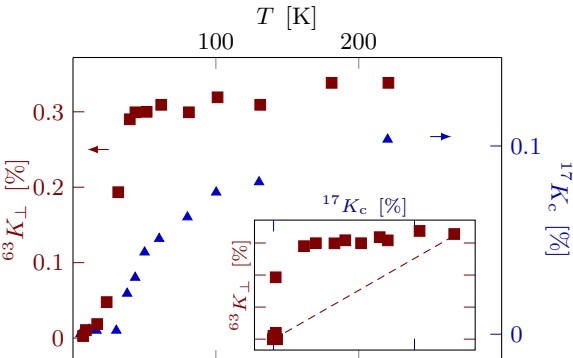

**Figure 5.** Failure of single fluid model for $La_{1.85}Sr_{0.15}CuO_4$, data from [40]. The temperature dependences of the shifts for $^{63}Cu$ and $^{17}O$ are rather different. Proportionality of the changes for $^{63}K_\perp$ and $^{17}K_c$, cf. (5), is violated, as indicated in the inset.

In a parallel set of experiments, the $HgBa_2CuO_{4+\delta}$ family of materials was investigated [26,27,41]. These materials are tetragonal and have a single $CuO_2$ plane similar to the $La_{2-x}Sr_xCuO_4$ family. Thus, any multi-component behavior must result from the plane (and, e.g., any influence from other spin components such as in the CuO-chains for the $YBa_2Cu_3O_{6+y}$ family can be excluded). Furthermore, the widely held belief was that, since doping occurs further away from the $CuO_2$ plane, these systems must be much more homogeneous compared to $La_{2-x}Sr_xCuO_4$. However, in first experiments on high quality single crystals, it was found that, while the Hg NMR shows very narrow lines, the Cu nuclei in the plane have a similar quadrupolar broadening of the satellite transitions due to charge variations as in $La_{1.85}Sr_{0.15}CuO_4$ [26], which was ascribed to charge density order only very recently [55] (but may not explain all doping and temperature dependences observed with NMR).

An important difference of the $HgBa_2CuO_{4+\delta}$ family of cuprates compared to $YBa_2Cu_3O_{6+y}$ and $La_{2-x}Sr_xCuO_4$ is the presence of a temperature dependent shift also for $c \parallel B_0$ for Cu. By comparing the shifts for Cu and both directions of the field, it became obvious that, for an underdoped system, the changes in shift with temperature in both directions were not proportional to each other, while the optimally doped material did show proportional changes [41]. On the other hand, Hg NMR showed that also the optimally doped system is not a single fluid [56]. In order to understand this somewhat mysterious behavior, more systems were studied [27]. With the new samples, it became apparent that there was a third shift component that had a different Cu shift anisotropy, easily recognizable for underdoped and overdoped samples. However, while this component was temperature independent at higher temperatures, the temperature where it rapidly vanished was different from $T_c$ (only for the underdoped sample investigated earlier is this temperature close to its $T_c = 74$ K). Furthermore, this component changes sign at optimal doping so that it is absent in the data. This new component can be recognized easily in a shift–shift plot, where $K_\perp$ is plotted against $K_\parallel$ (cf. Figure 6).

All of the data could be analyzed with these three susceptibilities, and it was argued that one of the susceptibilities could be due to the coupling of two spin components (cf. (7)). Interestingly, the Fermi liquid-like component shares the same anisotropy as the pseudogap component [27].

While all of these experiments made it clear that the approach used up to now that interprets the cuprate NMR shifts fails, and despite some support from the phenomenological theory [57] that used the earlier findings within a generalized two-fluid scenario, the single-fluid approach is still

being adopted (e.g., [58,59]), perhaps since a unifying, new phenomenology for the NMR data analysis is missing. With the new understanding of the cuprate phase diagram in terms of NMR that has two charge components (at Cu and O) [34], we decided to take a closer look at all available Cu shift data, which led us eventually to introduce a new, contrasting NMR shift phenomenology that will be discussed below.

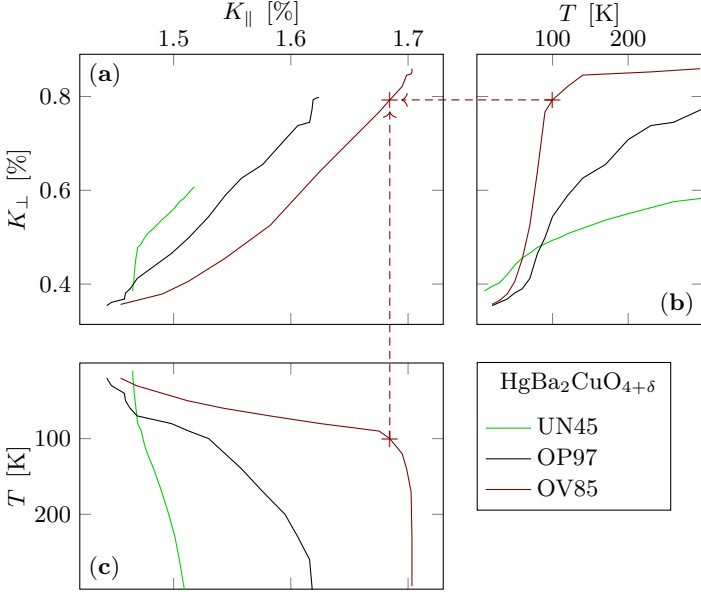

**Figure 6.** Failure of single component model for HgBa$_2$CuO$_{4+\delta}$ [27,41]. The temperature dependences of the $^{63}$Cu shifts with the magnetic field perpendicular and parallel to the *c*-axis are shown in panels (**b**) and (**c**), respectively, for three different doping levels: UD45, underdoped system with $T_c = 45$ K; OP97, optimally doped system with $T_c = 97$ K; OV85, overdoped system with $T_c = 85$ K. In (**a**), both shifts are plotted against each other with *T* as an implicit parameter, as indicated by the dashed lines. The changes in both shifts ($K_\perp$ and $K_\parallel$) are proportional to each other only at higher temperatures.

## 5. Contrasting Cu Shift Phenomenology

We collected data from the literature where Cu shifts were reported with the magnetic field parallel and perpendicular to the *c*-axis of the crystal. The shift reference was checked and changed to the diamagnetic part if necessary, as described in Section 2. The individual plots of shift vs. temperature can be found in the Supplementary Materials as well as the references collected in the Appendix A, where the reader can also find information about the changes applied to the data.

A first overview is shown in Figure 7 where we use the plot already discussed in more detail in Figure 6 [27,41]. It has $K_\perp$ as a function of $K_\parallel$ with temperature as an implicit parameter. Note that both axes have the same scale.

We would like to summarize what we think are key observations in Figure 7, before we discuss these points in greater detail below:

**A**　Common $K_\perp (T \to 0)$: all shifts for $c \perp B_0$ meet at a similar low-temperature shift point $K_\perp (T \to 0)$ $\approx 0.35\%$, dashed red line.

**B**　Large spread for $K_\parallel (T \to 0)$: for $c \parallel B_0$ different families have rather different low-temperature shifts $K_\parallel (T \to 0) \approx 1.2$–$2.0\%$.

**C**　Isotropic shift line: for overdoped systems and high temperatures, many points make up a line with slope $\Delta K_\perp / \Delta K_\parallel \approx 1$, as expected from isotropic hyperfine coupling (dashed diagonal line). There may be other isotropic lines for other families, and one of them is indicated by the dotted line.

**D**　New orbital shifts: the isotropic shift line and the common $K_\perp(T \to 0)$ intersect at $K_{L,\perp} \approx 0.35\%$ and $K_{L,\parallel} \approx 1.08\%$, which defines an orbital shift pair.

**E**　Shift triangle: practically all temperature dependent shifts are found below the main isotropic shift line, i.e., $K_\perp(T) \lesssim K_\parallel(T)$ for almost all cuprates, very different from what was assumed so far.

**F**　Characteristic slopes: a few typical slopes dominate the figure, $\Delta K_\perp / \Delta K_\parallel \approx 2.5$ and $\Delta K_\perp / \Delta K_\parallel \gtrsim 10$.

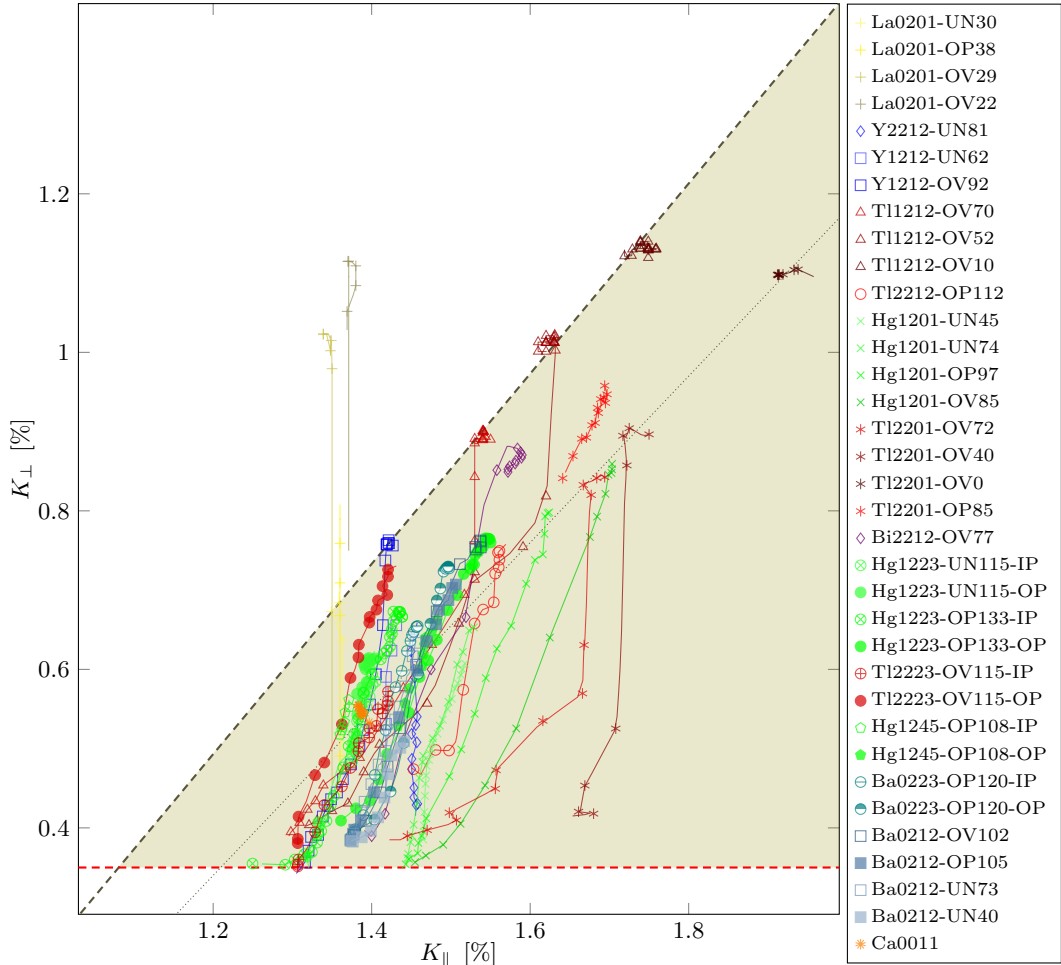

**Figure 7.** Planar Cu (total) shifts for $c \perp B_0$ ($K_\perp$) and $c \parallel B_0$ ($K_\parallel$) plotted against each other with temperature as an implicit parameter (similar to the plot in Figure 6). For the discussion of the plot, see (A) to (F) in the text.

We would like to discuss these six points somewhat further before we draw conclusions about the electronic liquid.

(A) The fact that $K_\perp$ reaches similar values at low temperatures independent from doping has been taken as the hallmark of spin singlet pairing. Slight differences in $K_\perp(T \to 0)$ could stem from variations in the orbital shift, the life-time of the electronic states, differences in macroscopic diamagnetism, quadrupole interaction (charge density variations), or even somewhat different field-dependent shifts. While partial diamagnetism may shift $K_\perp$ somewhat up at low temperatures, it appears to be a very reliable shift.

(B) Most systems have temperature dependent shifts for $c \parallel B_0$ ($K_\parallel(T)$), contrary to the systems analyzed early on. In fact, these shifts can be rather large, which would require very different hyperfine

scenarios assuming the traditional shift phenomenology. However, at low temperatures, these shifts do not converge to a similar $K_\parallel(T \to 0)$ as for $K_\perp$. Given the rather ubiquitous $CuO_2$ plane, it appears difficult to understand the differences only in terms of variable orbital shifts.

(C) A great number of shifts for overdoped systems above $T_c$ ($K_{\perp,\parallel}(T > T_c)$) define a line with a slope of nearly 1, i.e., $\Delta_x K_\perp / \Delta_x K_\parallel \approx 1$, where we denote with $\Delta_x$ the change with respect to doping. Changing temperature has a very different effect than changing doping. Apparently, this isotropic shift line varies somewhat between families (dotted and dashed lines in Figure 8b). Since most of the $K_\alpha(T > T_c)$ involve the overdoped, temperature independent shift range, one might be inclined to assign this line to a shift due to a simple (Fermi) liquid that reigns in the overdoped systems at high temperatures and to which the nuclei couple predominantly with an almost isotropic hyperfine coefficient. We also note that there is a parallel line to the dominant isotropic line: after a drop at $T_c$ as the temperature is lowered, the shifts disappear as a function of temperature (not doping) in an isotropic fashion for Tl1212 (cf. Figure 8a).

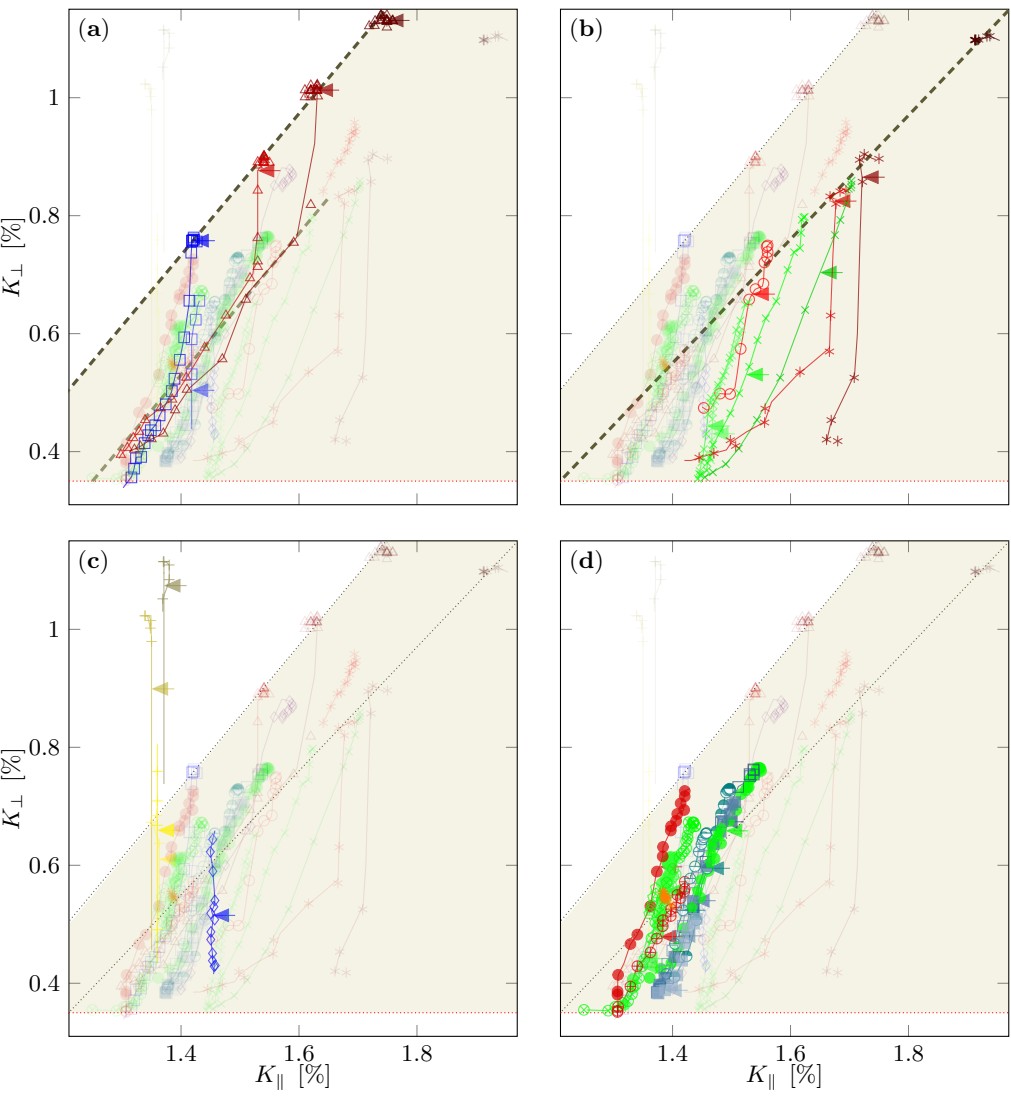

**Figure 8.** Accentuation of data in Figure 7: two nearly isotropic lines emerge with a slope $\Delta K_\perp / \Delta K_\parallel \approx 1.2$ (**a**) and $\Delta K_\perp / \Delta K_\parallel \approx 1.05$ (**b**). Note that, after initial drop at $T_c$ as the temperature is lowered, the shifts fall again according to the high-temperature nearly isotropic lines in (**a**). $K_\parallel$ for La0201 and Y2212 systems does not depend on temperature (**c**). The triple layer systems, the quintuple layer Hg1245, and the infinite layer Ca0011 have a common small shift window (**d**). Arrows indicate $T_c$.

(D) The isotropic shift lines intersect $K_\perp(T \to 0)$ and define orbital shifts $K_{L,\parallel}$. This may indicate slight differences in the orbital shift for different families. The new values for the orbital shift are much closer to what is expected from first principle calculations that predict an orbital shift anisotropy of about 2.4 [36].

(E) Another striking observation in Figure 7 is the fact that almost all experimental shifts are below the highest isotropic line (or the isotropic line for the family), i.e., $K_\perp \lesssim K_\parallel$ at all temperatures. If the diagonal lines define a Fermi liquid of carriers with isotropic coupling the action of the gaps, as the temperature is lowered, leads to a stronger decrease of $K_\perp$ than $K_\parallel$.

(F) When looking at Figure 7, certain other straight line segments can be identified. Next to the isotropic shift line, one recognizes a somewhat steeper slope that was described first with the HgBa$_2$CuO$_{4+\delta}$ family of cuprates [27,41]. It was assumed that it is caused by the anisotropy of a hyperfine coupling coefficient. The offsets between the lines were believed to be due to another shift component that disappears below a characteristic temperature $T_0 \neq T_c$. For YBa$_2$Cu$_3$O$_7$ (Y1212), the special slope appears well below $T_c$, while, for other samples (e.g., the HgBa$_2$CuO$_{4+\delta}$ (Hg1201) family), it appears above and below $T_c$, and this is shifted by doping. For a given family, the low-temperature shift values meet at the same point, while there is an offset between families (of the size of what is observed even between members of a single family at higher temperatures). Another much steeper slope with $\Delta K_\parallel \approx 0$ can be found as well (cf. Figure 8c). It is important for underdoped YBa$_2$Cu$_3$O$_{6+y}$ (Y1212), La$_{2-x}$Sr$_x$CuO$_4$ (La0201), and some Tl compounds in some range of temperatures.

## 6. Discussion

A number of systems [27,40,41,51] were shown to fail for a single fluid model explanation, as summarized in Section 4. In particular, one of the benchmark systems (YBa$_2$Cu$_4$O$_8$) was found to be just at the brink of failing for the single fluid picture, as application of modest pressure revealed [54]. Most of the evidence came from contrasting temperature dependences of the shifts for different nuclei, but also just for a single nucleus and different orientations of the external field [27,41,51]. Since the Cu shift data appear quite reliable, as discussed above, we gathered available data and applied the same shift referencing. The data were then plotted in Figure 7 as $K_\perp$ vs. $K_\parallel$ (with temperature as an implicit parameter).

By inspecting Figure 7, we pointed to a number of rather reliable facts that lead to our new shift phenomenology for the cuprates. The first such fact is that the low-temperature shifts for $c \perp B_0$ converge to the same shift for all materials, which must determine the orbital shift ($K_{L,\perp} \approx 0.35\%$), perhaps the only assumption that survives the old picture. Quite the opposite, the shifts for $c \parallel B_0$ do not converge towards a common value. While it is possible that the orbital shift in this direction does depend on the family of materials, there must be an unexplained, residual shift left at the lowest temperatures. Part of the evidence comes from the plot itself (as already mentioned), but also first principle calculations predicted an orbital shift anisotropy of $K_{L,\parallel}/K_{L,\perp} \approx 2.4$, a number that appears rather reliable also on general grounds [36], and hints at an orbital term $K_{L,\parallel} \approx 0.85\%$, much closer to what we determine ($K_{L,\parallel} \approx 1.05\%$) from Figure 7. We know from the HgBa$_2$CuO$_{4+\delta}$ family, investigated with great care earlier [27,41], that this additional shift is probably present at Hg, as well as planar O, which favors a spin contribution.

Another very reliable observation is what we call the isotropic shift line, a line with a slope of nearly one ($\Delta K_\perp/\Delta K_\parallel \approx 1$) in Figure 7. Probably, there are few such lines with slightly different slopes for different families (cf. also Figure 8b). These lines clearly show that the extent of parallel and perpendicular shifts must be similar, very different from earlier conclusions, but in support of what we just stated above. This most likely calls for a much larger isotropic hyperfine coefficient $\tilde{B} \gg B$, again contrary to what has been assumed in the early phenomenology. The nearly isotropic shift lines also define an orbital shift $K_{L,\parallel}$ together with $K_{L,\perp}$ that is much closer to what is expected from the discussion above. Perhaps slightly different points of intersection of the nearly isotropic shift lines with a common $K_{L,\perp}$ define slightly different orbital shifts for different families.

Another remarkable observation is that almost all shift data are found below the diagonal (nearly isotropic) shift line, i.e., when lowering the temperature so that either the pseudogap or the superconducting gap sets in $K_{s,\perp}$ falls faster than $K_{s,\parallel}$. Furthermore, this happens interestingly enough along a small number of rather fixed slopes $\Delta K_\perp / \Delta K_\parallel$. This must point to special properties of the fluid.

How can we understand the findings with a minimal amount of assumptions that do not rest on particular theoretical pictures? Based on the chemical bonding, we expect an isotropic term that involves the Cu $4s$ orbital (in various possible ways). If the related hyperfine coefficient $\tilde{B}$ is sufficiently large (we just use $\tilde{B}$ instead of $4\tilde{B}$), it explains why any spin density $\langle S_B \rangle$ creates the nearly isotropic shift line. We write,

$$K_{s,\alpha} = \frac{A_\alpha + \tilde{B}}{\gamma_e \hbar} \langle S_B \rangle, \tag{8}$$

where we added an anisotropic term $|A_\alpha| \ll \tilde{B}$ so that $K_{s,\perp}/K_{s,\parallel} \approx 1$ since the isotropic lines deviate only slightly from slope 1 and since the local chemistry demands an involvement of the $3d(x^2 - y^2)$ orbital. Its bare onsite core polarization, dipolar, and spin-orbit contributions are quite well known and lead to an overall anisotropy $|A_\parallel / A_\perp| \gtrsim 6$ with a negative $A_\parallel$ and an almost vanishing $A_\perp$ [35,44]. Thus, spin density in this orbital will predominantly change $K_{s,\parallel}$, i.e., it will lead to a shift change that is nearly on a horizontal line in Figure 7. Without having to speculate about the details of the electronic fluid(s), one would then guess that what is needed in order to explain the temperature dependences of the data in Figure 7 is a change of spin density in the $3d(x^2 - y^2)$ orbital leading to $A_\parallel \Delta \langle S_A \rangle$ (nearly horizontal change) and a concommittant change for the isotropic part $\tilde{B} \Delta \langle S_B \rangle$ (cf. Figure 9). The special slopes discussed above would then tell us about the allowed ratios $\tilde{B} \Delta \langle S_B \rangle (T) / A_\parallel \Delta \langle S_A \rangle (T)$ that make up the few special slopes in Figure 7.

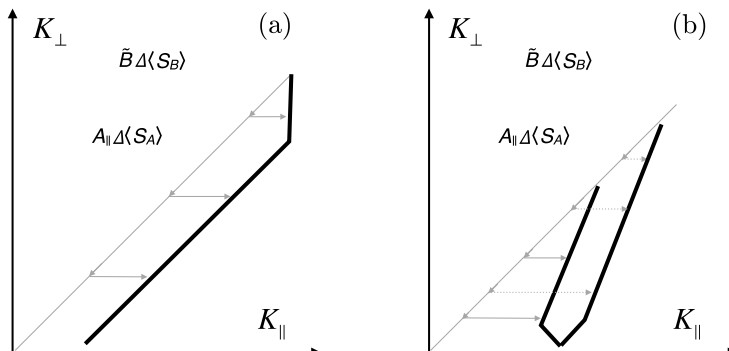

**Figure 9.** Construction of special slopes in terms of two spin components $\langle S_A \rangle$ and $\langle S_B \rangle$ that lead to nearly horizontal and diagonal shifts in Figure 7, respectively. (**a**) Slope predominantly parallel to the isotropic shift line; (**b**) Main slope of about 2.4.

In the above model and in view of the figures above, decreasing the temperature leads to basically two different behaviors.

(1) Systems that undergo a superconducting transition without a preceding shift decrease (from the action of the pseudogap) rapidly lose (with the large slope) a certain amount of predominantly $K_{s,\perp}$. As the temperature decreases further, both shifts fall parallel to the isotropic line. This indicates a certain, rapid decrease in $A_\parallel \langle S_A \rangle$ that creates a similar shift change as $\tilde{B} \langle S_B \rangle$, followed by a change in $\langle S_B \rangle$ only.

(2) Systems that lose shifts by the action of the pseudogap as the temperature is lowered, before the shift change from the superconducting gap that appears, behaves differently. Here, the slope is about 2.4 with which the shifts break away from the isotropic line. This slope is not interrupted

by crossing $T_c$. Only at very low temperatures do the changes in shifts break away from the special slope according to the following phenomenology: after $\langle S_A \rangle$ is fully established, $\langle S_B \rangle$ still falls for the overdoped systems; for the underdoped systems, the change in $\langle S_A \rangle$ is still not completed when that in $\langle S_B \rangle$ is exhausted. This scenario is apparent for the $HgBa_2CuO_{4+\delta}$ family, and, for other systems, the typical deviations from the linear slopes hint at the same behavior. For optimal doping, the shifts appear to follow the high-temperature slope down to the lowest values.

At the lowest temperatures, there must be negative spin density left in the $3d(x^2 - y^2)$ orbital, which may be difficult to detect with uniform susceptibility measurements that suffer from the Meissner response and notorious contributions from Curie defects at low temperatures.

We would like to note that, in the extensive analysis of the failure of the single fluid scenario for $La_{1.85}Sr_{0.15}CuO_4$, it was already reported [40] that the coupling of the Cu nucleus to the Fermi liquid-like component was 8.6 times larger than its coupling to the other component, and that the planar oxygen was found to only weakly couple to the Fermi liquid-like component. Thus, for systems where the temperature dependence of the Cu shift is not dominated by the Fermi liquid-like component, Cu and O should show a similar temperature dependence. This explains why the single fluid behavior was detected with some of the materials early on. The action of the superconducting gap with its pronounced, sudden drop in shift at $T_c$ is not always seen in the Cu data (cf. Figure 7), but hardly for planar oxygen. This explains why the Fermi liquid-like component as a second shift component can easily be missed.

Finally, we would like to discuss the findings in view of the NMR phase diagram [34] that was suggested recently. We do not observe an immediate trend relating $T_c$ to the NMR shifts in Figure 7. However, there appears to be a subtle connection to the charge distribution between planar Cu and O measured by NMR [33]. The degree of covalency of the planar Cu-O bond, i.e., the hole distribution between Cu and O that is set by material chemistry and is related to the charge transfer gap, was shown to set the maximum $T_c$ and superfluid density [33,34]. In Figure 8c, the $La_{2-x}Sr_xCuO_4$ family, having the lowest O hole content and the lowest maximum $T_c$, is limited to a very narrow window of $K_\parallel = 1.35 - 1.40\%$ such that doping and increasing temperature only affect an increase in $K_\perp$. The Y-based materials with higher O hole content already in the parent material and higher $T_{c,max}$ cover a wider range of $K_\parallel = 1.30 - 1.46\%$ while changes in $K_\perp$ are dominating. Finally, the Hg, Tl and Bi-based materials that show the highest O hole contents and correspondingly highest $T_c$ span the entire range of observed $K_\parallel$ as a function of doping and temperature, where, interestingly, the range of observed $K_\parallel$ decreases with an increase of adjacent $CuO_2$ layers. We note that these materials, particularly the single-layer materials (Hg1201, Tl2201), show the greatest spread, $K_\parallel = 1.40 - 1.95\%$, and, for high temperatures, they fall on the isotropic shift line ($\Delta_x K_\perp / \Delta_x K_\parallel = 1.05$). The double-layer materials of Tl1212 span a slightly smaller range, $K_\parallel = 1.30 - 1.78\%$, and, for high temperatures, they fall on the nearly isotropic shift line ($\Delta_x K_\perp / \Delta_x K_\parallel = 1.2$) together with double-layer $YBa_2Cu_3O_{6+y}$. The triple (Hg1223, Tl2223) and quintuple (Hg1245) layer materials are limited to $K_\parallel = 1.25 - 1.55\%$, Figure 8d.

Therefore, while we do not find an obvious relation of the shift phenomenology to $T_c$, materials with increased O hole content (at the expense of that at Cu) do show much larger isotropic shifts at high temperatures.

## 7. Conclusions

Based on literature shift data for planar Cu, we developed a contrasting shift phenomenology for the cuprate superconductors, which is quite different from the early view. For example, the data show that a different hyperfine scenario must be invoked since there is obviously a large isotropic shift present in the cuprates. We also find a new orbital shift for $c \parallel B_0$ that is in much better agreement with predictions from first principle calculations compared to the old picture. We discussed how the pseudogap and the superconducting gap change the shifts when the temperature is lowered. This proceeds according to a few rules that must be explained by theory. It appears that there is residual spin shift for $K_\parallel$ at the lowest temperature, which points to a localization of spin in the $3d(x^2 - y^2)$

orbital. The new scenario is in agreement with all shift data that concluded on single as well as multiple components earlier. Clearly, a single temperature dependent spin component cannot explain the cuprate shifts. A simple model is presented that is based on the most fundamental findings.

**Supplementary Materials:** The following is available online at http://www.mdpi.com/2410-3896/2/2/16/s1, Figures: shift data for each group.

**Acknowledgments:** We are thankful for discussions with Oleg P. Sushkov, Grant V. M. Williams, Dirk K. Morr, Steven Reichardt, Robin Guehne, and Jan Zaanen, and acknowledge financial support from the University of Leipzig.

**Author Contributions:** Jonas Kohlrautz and Michael Jurkutat collected and extracted the literature data; and Jürgen Haase, Jonas Kohlrautz, and Michael Jurkutat analyzed the data and wrote the paper. All authors have read and approved the final manuscript.

**Conflicts of Interest:** The authors declare no conflict of interest.

## Appendix A. Literature Data

In Table A1, we list all materials from which shift data were extracted, giving the abbreviation [60] and corresponding stoichiometric formula, the literature reference and noting shift corrections that we performed.

In all works, the respective authors appear to have corrected for the second-order quadrupole effect in the determination of the $K_\perp$, which was usually done assuming a temperature-independent quadrupole splitting. This is justified, since $\nu_Q$ is not expected to change significantly with temperature and the reported studies of $\nu_Q(T)$ reveal only slight temperature dependences. Correspondingly, changes in the second order effect are negligible for the determined shifts.

Furthermore, the temperature dependent data highlighted for each group can be found in Supplementary Materials. In all plots, linear interpolation was applied. For completeness, extracted shift data listed but not presented in Section 5 (cf. Table A1) are shown in Figure A1.

**Table A1.** List of included data with material abbreviation, full stoichiometric formula and reference for the original data with footnotes explained below the table. Data are not corrected for diamagnetic contributions lacking internal NMR references.

| Symbol | System | Reference |
|---|---|---|
| La0201 | $La_{2-x}Sr_xCuO_4$ | [61] [2] |
| Y1212 | $YBa_2Cu_3O_{6+x}$ | [42,43] [2] |
| Y2212 | $YBa_2Cu_4O_8$ | [47] [2] |
| Tl1212 | $TlSr_2CaCu_2O_{7-\delta}$ | [62] [2] |
| Tl2201 | $Tl_2Ba_2CuO_{6+y}$ | [63,64] [2] |
| Tl2212 | $Tl_2Ba_2CaCu_2O_{8-\delta}$ | [65] [3] |
| Tl2223 | $Tl_2Ba_2Ca_2Cu_3O_{10}$ | [66] [2] |
| Hg1201 | $HgBa_2CuO_{4+\delta}$ | [27] |
| Hg1223 | $HgBa_2Ca_2Cu_3O_{8+\delta}$ | [67,68] [2] |
| Hg1245 | $HgBa_2Ca_4Cu_5O_y$ | [69] |
| Ba0223 | $Ba_2Ca_2Cu_3O_6(F,O)_2$ | [70] [2] |
| Ba0212 | $Ba_2CaCu_2O_6(F,O)_2$ | [70] [2] |
| Bi2212 | $Bi_2Sr_2CaCu_2O_8$ | [71] [2] |
| Ca0011 | $Ca_{0.85}Sr_{0.15}CuO_2$ | [72] [1] |
| (Y,Pr)1212 | $Y_{1-x}Pr_xBa_2Cu_3O_7$ | [73] [2,5] |
| Tl2223 | $Tl_2Ba_2Ca_2Cu_3O_{10}$ | [74] [2,4] |
| Tl2223 | $Tl_2Ba_2Ca_2Cu_3O_{10}$ | [75] [1,4] |
| (Tl,Pb)1212 | $Tl_{0.5}Pb_{0.5}Sr_2Ca_{1-x}Y_xCu_2O_{7-\delta}$ | [76] [2,4] |
| (Bi,Pb)2223 | $Bi_{1.6}Pb_{0.4}Sr_2Ca_2Cu_3O_{10}$ | [77] [2,4] |

[1] shift values were increased by 0.15% to account for explicitly stated CuCl (or CuSO$_4$) reference; [2] no clearly stated reference, shift values were increased by 0.15% to account for an assumed CuCl reference; [3] shift values were increased by 0.38% to account for an explicitly stated reference to metallic Cu; [4] shift data excluded from Section 5 owing to unclear spectral assignment due to very broad lines and/or spectral overlap and/or contradiction to other literature data; [5] shift data excluded from Section 5 due to effect of 4f magnetism of rare earth atoms in a charge reservoir layer.

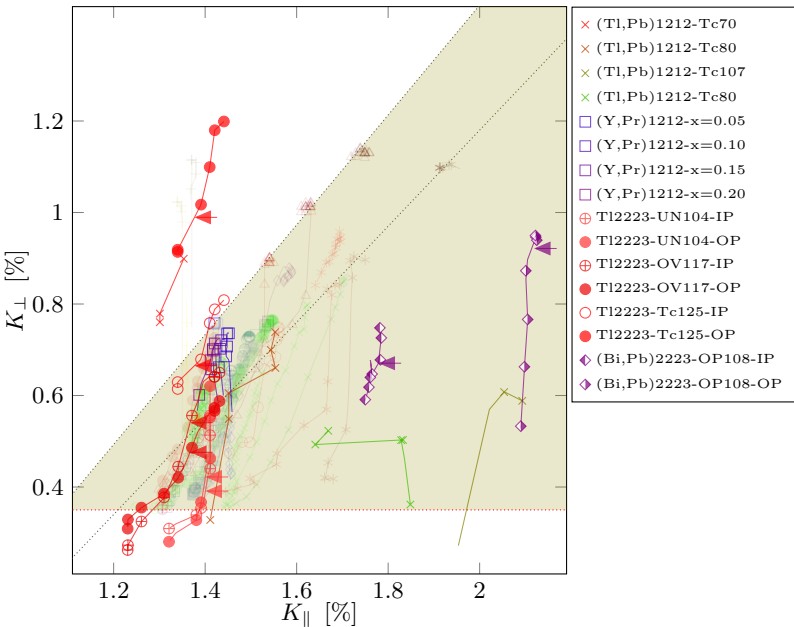

**Figure A1.** Additional data sets, extracted and listed in Table A1, which were not presented in Section 5. For comparison, data from Figures 7 and 8 are also shown as faded here. Arrows indicate $T_c$.

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
