# Peer review of "Contrasting Phenomenology of NMR Shifts in Cuprate Superconductors"

_condensedmatter, doi:10.3390/condmat2020016_

Reviewer 1 Report

The paper provides an exzellent and timely review over the state of the Art in NMR Research for high temperature superconducting copper Oxides, and summarizes all Data currently available on NMR in cuprate Oxides. The authors discover an unknown Relation between hole Doping and Show that Planes

 and out of plane contributions differ substantially. Since no theortical Input is given, the results are free of misinterpretations.

Author Response

Thank you for reviewing our manuscript and endorsing publication.

Reviewer 2 Report

The authors have tried to make a systematic overview of existing NMR shift data in order to support the two electronic component model for high-Tc superconductors in contrast to previously favoured single electronic fluid model. They have carefully collected ample literature data for single crystals or oriented samples where Cu NMR shifts were reported for two orientations. The reported shifts are corrected for the reference where necessary. The main result of the manuscript is that almost all historically reported shift data are consistent with the two electronic component model, even those which were previously taken as the proof for the validity of single component model.

This comprehensive study is worth publishing after addressing the following points:

1)    The single component model was widely accepted in nineties, but in the last decade it seems that two-component model is accepted by most of the researchers in the field. It is my opinion, but it is also strongly supported by the presentation in the manuscript. The cited literature supporting single component model is dated between 1989 and 1997. Is it true that researches abandoned the single component model in the 21st century? If yes, then the present manuscript represents just one more confirmation of the accepted model. If not, then the authors should mention current literature supporting the single component model, and explain why they think this literature makes wrong conclusions.

2)    In support to two-component model, the authors cite several papers (Refs. 26-28, 40, 41, 43) written by them. However, some other papers supporting the model are not mentioned. The conclusion would be much stronger if e.g. [Barzykin and Pines, Adv. Phys. 2009], [Crocker et al. PRB 2011], and other were properly cited.

3)    Some data were excluded from Fig. 7. For example, there are three references for Tl2223 ([52],[59], [60]), but only the one which fits into the “triangular story” is plotted in Fig. 7. Inclusion of these data would weaken the points (A) and (E) in discussion on contrasting Cu shift phenomenology. The authors should better elaborate why these data are excluded, not only because they contradict to the other data.

4)    Line 230: How could quadrupolar broadening due to charge variations affect the width of the central line? Moreover, according to [42] the charge density order vanishes at dopings below optimal, while the NMR linewidths of planar Cu are large at all dopings. The original explanation for the broad central lines, given by authors in [41], (distribution of dopings) is more reasonable.

5)    Talking about ref. [42] it should be cited as W. Tabis et al, since it is authored by 17 authors.

6)    Prossible typo in line 366: the sentence is not logically constructed.

7)    Probably a typo: The second paragraph of conclusion (lines 432-434) doesn’t seem to be a part of the manuscript.

Not necessary, but in line with point 1) above, the authors might wish to comment a recent paper by Chen in Nat. Commun. 14986, which was published after they submitted the manuscript.

Author Response

We thank the Reviewer 2 for bringing up various helpful thoughts and we made changes to the paper accordingly. Here, we would like to address the Reviewers points, one by one, in a more general way.

>The authors have tried to make a systematic overview of existing NMR shift data in order to support the two electronic >component model for high-Tc superconductors in contrast to previously favoured single electronic fluid model. They have >carefully collected ample literature data for single crystals or oriented samples where Cu NMR shifts were reported for two >orientations. The reported shifts are corrected for the reference where necessary. The main result of the manuscript is that >almost all historically reported shift data are consistent with the two electronic component model, even those which were >previously taken as the proof for the validity of single component model.

>This comprehensive study is worth publishing after addressing the following points:

>1)    The single component model was widely accepted in nineties, but in the last decade it seems that two-component model >is accepted by most of the researchers in the field. It is my opinion, but it is also strongly supported by the presentation in the >manuscript. The cited literature supporting single component model is dated between 1989 and 1997. Is it true that researches >abandoned the single component model in the 21st century? If yes, then the present manuscript represents just one more >confirmation of the accepted model. If not, then the authors should mention current literature supporting the single component >model, and explain why they think this literature makes wrong conclusions.

In 2007, in the Schrieffer Book on high temperature superconductivity, one of the pioneers of magnetic resonance also in the field of cuprate superconductivity, C.P. Slichter, mentions that the hitherto adopted single-fluid picture might not hold [Slichter:2007]. Then, in 2009, together with him, some of us published the first 100% proof of the failure of the single fluid scenario for one system, while it remained unclear, back then, why the other two compounds investigated in 1991 and 1994 supported the single fluid picture. J.H. knows personally from long discussions with supporters of the single-fluid scenario, e.g., M. Rice, D. Pines, and many others, that they still  held onto the single fluid picture thinking that LaSrCuO, which  we discussed in the 2009 paper, might be an unfortunate exception. The book by Walstedt, another pioneer in the field, in 2008 also supports the single-fluid picture [Walstedt:2008]. David Pines, was converted to the new view and the Reviewer points correctly to his paper where V. Barzykin and D. Pines suggest a phenomenological model for two-component physics [Barzykin:2009]. Still, in 2013 in a Phys. Rev. Lett. by Mounce et al. [Mounce:2013] the single-component view is endorsed, while not proven. We also conclude from the new reference Reviewer 2 brought to our attention [Chen:2017] that they follow the single-fluid view, even today, but do not prove or disprove it. In this sense, we do not share the view of the Reviewer 2 that the single-component picture is a thing of the 1990ies. Of course, we are by no means able to assess what the more general thinking is among theorists, but it is certainly very diverse.

More importantly, we do not adopt a particular model in our new manuscript. Our phenomenology does not make any assumptions about the number of components of the fluid. Instead, we only want to point out what a more complete set of data looks like, and we point to simple relationships that must contain hints for theory. Whether the data can be explained by a special single fluid or whether one needs three of four components, we don’t know and don’t want to speculate about.

[Slichter:2007]C. P. Slichter, in Handb. High-Temperature Supercond., edited by J. R. Schrieffer and

J. S. Brooks (Springer, New York, 2007), pp. 215–256.

[Walstedt 2008]  Russell E. Walstedt, The NMR Probe of High-Tc Materials, Springer 2008

[Barzykin:2009]V. Barzykin and D. Pines, Advances in Physics 58, 1 (2009).

[Mounce:2013]A. M. Mounce, S. Oh, J. A. Lee, W. P. Halperin, A. P. Reyes, P. L. Kuhns, M. K. Chan, C. Dorow, L. Ji, D. Xia, X. Zhao, and M. Greven, Phys. Rev. Lett. 111, 187003 (2013).

[Chen:2017]X. Chen, J. P. F. LeBlanc and E. Gull, and, Nature Communications 8, 1 (2017).

>2)    In support to two-component model, the authors cite several papers (Refs. 26-28, 40, 41, 43) written by them. However, >some other papers supporting the model are not mentioned. The conclusion would be much stronger if e.g. [Barzykin and >Pines, Adv. Phys. 2009], [Crocker et al. PRB 2011], and other were properly cited.

Yes, we did not mention all papers that had problems with a single-fluid picture, but we mention close to all of those papers in our other papers on the failure of the single-fluid picture. Nevertheless, we included one of the papers the Reviewer points to. See Summary of changes for details.

>3)    Some data were excluded from Fig. 7. For example, there are three references for Tl2223 ([52],[59], [60]), but only the >one which fits into the “triangular story” is plotted in Fig. 7. Inclusion of these data would weaken the points (A) and (E) in >discussion on contrasting Cu shift phenomenology. The authors should better elaborate why these data are excluded, not only >because they contradict to the other data.

Yes, a very few data sets were omitted from Fig. 7, but they are shown later in another plot (A1). Those that we omitted did raise concerns with us about proper referencing, not because they disturb the beauty of the triangular story. For instance, Piskunov et al. 1998 [60] in fact acknowledge that their precision was inferior compared to Zheng et al. 1996 [52].

>4)    Line 230: How could quadrupolar broadening due to charge variations affect the width of the central line? Moreover, >according to [42] the charge density order vanishes at dopings below optimal, while the NMR linewidths of planar Cu are large >at all dopings. The original explanation for the broad central lines, given by authors in [41], (distribution of dopings) is more >reasonable.

Good point. We changed the wording and say that it concerns the satellite transitions, this should alleviate the confusion. In addition, yes, the reviewer is correct that the broadening found with NMR is not (yet) in agreement with what was reported. We are happy to follow the suggestion of the reviewer and change that.

>5)    Talking about ref. [42] it should be cited as W. Tabis et al, since it is authored by 17 authors.

Thank you, we changed that.

>6)    Prossible typo in line 366: the sentence is not logically constructed.

We improved that.

>7)    Probably a typo: The second paragraph of conclusion (lines 432-434) doesn’t seem to be a part of the manuscript.

Thank you for the comment, our mistake.

>Not necessary, but in line with point 1) above, the authors might wish to comment a recent paper by Chen in Nat. Commun. >14986, which was published after they submitted the manuscript.

Chen et al. actually employ a single band view (cf. eq. (4) in Chen:2017 and eq. (6) in our manuscript) when translating their results into NMR shifts and we included the reference correspondingly. Beyond that, we cannot judge the implications of the theoretical work by Chen and co-workers for our manuscript.

Reviewer 3 Report

Jurgen Haase and coworkers present a nice extensive overview on the phenomenology of the NMR shift in high Tc superconducting cuprates, concentrating on Cu(2) shift where more reliable data are available. Their main finding is that a single-fluid scenario, widely accepted at the very beginning of the high Tc era, has to be revisited. This observation is not by itself novel and the need to go beyond this single-band model was already pointed out long ago by the Grenoble NMR group (see for example T.Auler et al., Physica C 313, 255 (1999) and references therein).  However, what is important here is that the analysis of the NMR shift presented does not rely on a specific model, making the observations by Haase et al. more solid. Another important remark is that there is a clear increase in the isotropic part of the shift with increasing hole content and that, while at high T the Cu(2) shift tends to become more isotropic, at low T K_\perp decreases faster possibly due to a change of the spin density on 3d(x^2-y^2) orbitals. These and other relevant points are clearly summarized at Sect.5.

The manuscript is well written, the analysis appears to be solid and the findings can be very important for the understanding of the cuprates. So the manuscript certainly deserves to be published. I have only two concerns that the authors should consider before the manuscript is finally accepted for publication:

a)      For 63Cu K_\perp the quadrupolar corrections are sizeable as the authors point out. One may wonder if the temperature dependence of the quadrupolar frequency was taken into account in the derivation of K_\perp or if it can be neglected. Notice that if there is a temperature dependence of the electron occupancy of 3d orbitals this would affect alos the quadrupolar frequency.

b)       The authors did a great job in collecting all data on Cu shifts, shown in Figs.7 and 8.  However, there are probably too many data point shown and some symbols can hardly be distinguished. Moreover, some of the symbols are really faint and can hardly be seen.  I would suggest the authors to make more figures besides Fig.7, where all data are summarized, and use larger symbols. One could consider making a figure for each cuprate family. 

Author Response

We thank the Reviewer 3 for insightful comments with deep understanding of the matter. Here is a more detailed response to the comments.

>Jurgen Haase and coworkers present a nice extensive overview on the phenomenology of the NMR shift in high Tc >superconducting cuprates, concentrating on Cu(2) shift where more reliable data are available. Their main finding is that a >single-fluid scenario, widely accepted at the very beginning of the high Tc era, has to be revisited. This observation is not by >itself novel and the need to go beyond this single-band model was already pointed out long ago by the Grenoble NMR group >(see for example T.Auler et al., Physica C 313, 255 (1999) and references therein).  However, what is important here is that >the analysis of the NMR shift presented does not rely on a specific model, making the observations by Haase et al. more solid. >Another important remark is that there is a clear increase in the isotropic part of the shift with increasing hole content and that, >while at high T the Cu(2) shift tends to become more isotropic, at low T K_\perp decreases faster possibly due to a change of >the spin density on 3d(x^2-y^2) orbitals. These and other relevant points are clearly summarized at Sect.5.

>The manuscript is well written, the analysis appears to be solid and the findings can be very important for the understanding of >the cuprates. So the manuscript certainly deserves to be published. I have only two concerns that the authors should consider >before the manuscript is finally accepted for publication:

>a)      For 63Cu K_\perp the quadrupolar corrections are sizeable as the authors point out. One may wonder if the temperature >dependence of the quadrupolar frequency was taken into account in the derivation of K_\perp or if it can be neglected. Notice >that if there is a temperature dependence of the electron occupancy of 3d orbitals this would affect also the quadrupolar >frequency.

All papers we inspected appear to have taken into account the second-order quadrupole effect, but subtracted a temperature-independent part only. We added a sentence in the manuscript to make sure the reader is aware of this point. For those papers that do give a temperature-dependent quadrupole term its effect on the shift appears to be negligible. With regard to the change in occupancy of the x^2-y^2 orbital: the Reviewer 3 is correct; we added a comment in the revised manuscript.

>b)       The authors did a great job in collecting all data on Cu shifts, shown in Figs.7 and 8.  However, there are probably too >many data point shown and some symbols can hardly be distinguished. Moreover, some of the symbols are really faint and >can hardly be seen.  I would suggest the authors to make more figures besides Fig.7, where all data are summarized, and use >larger symbols. One could consider making a figure for each cuprate family.

Yes, perhaps. So we added in the Supplementary a file where one can look at the individual families.